# Confirmation of Statin and Fibrate Use from Small-Volume Archived Plasma Samples by High-Throughput LC-MS/MS Method

**DOI:** 10.3390/ijms24097931

**Published:** 2023-04-27

**Authors:** Jennifer D. Kusovschi, Anna A. Ivanova, Michael S. Gardner, Robert W. McGarrah, William E. Kraus, Zsuzsanna Kuklenyik, James L. Pirkle, John R. Barr

**Affiliations:** 1Clinical Chemistry Branch, Division of Laboratory Sciences, National Center for Environmental Health, Centers for Disease Control and Prevention, Atlanta, GA 30341, USA; 2Duke Molecular Physiology Institute, Duke University School of Medicine, Duke University, Durham, NC 27701, USA

**Keywords:** mass spectrometry, LC-MS/MS, human plasma, medical records, statins, fibrates, cardiovascular disease

## Abstract

Designing studies for lipid-metabolism-related biomarker discovery is challenging because of the high prevalence of various statin and fibrate usage for lipid-lowering therapies. When the statin and fibrate use is determined based on self-reports, patient adherence to the prescribed statin dose regimen remains unknown. A potentially more accurate way to verify a patient’s medication adherence is by direct analytical measurements. Current analytical methods are prohibitive because of the limited panel of drugs per test and large sample volume requirement that is not available from archived samples. A 4-min-long method was developed for the detection of seven statins and three fibrates using 10 µL of plasma analyzed via reverse-phase liquid chromatography and tandem mass spectrometry. The method was applied to the analysis of 941 archived plasma samples collected from patients before cardiac catheterization. When statin use was self-reported, statins were detected in 78.6% of the samples. In the case of self-reported atorvastatin use, the agreement with detection was 90.2%. However, when no statin use was reported, 42.4% of the samples had detectable levels of statins, with a similar range of concentrations as the samples from the self-reported statin users. The method is highly applicable in population studies designed for biomarker discovery or diet and lifestyle intervention studies, where the accuracy of statin or fibrate use may strongly affect the statistical evaluation of the biomarker data.

## 1. Introduction

According to the World Health Organization, cardiovascular diseases (CVD) caused 17.9 million deaths in 2019, 32% of the total deaths worldwide. Atherosclerotic cardiovascular disease (ASCVD), also known as coronary artery disease (CAD), is caused by the buildup of cholesterol-containing plaques in the arteries, which leads to heart attack and stroke, the direct cause of 85% of CVD-related deaths [1]. Primary risk assessment for CVD is based on lipid profile tests and consideration of other risk factors such as race, sex, age, obesity, diabetes, hypertension, and history of smoking [2]. Numerous epidemiological studies suggest that low-density lipoprotein (LDL) particles are the main cause of atherosclerosis [3,4,5,6].

Those at risk for ASCVD are often prescribed lipid-lowering medication, such as statins and fibrates, as a preventative measure. Statins are a class of drugs that function by competitively blocking the active site of β-Hydroxy β-Methylglutaryl-Coenzyme A (HMG-CoA). This inhibition causes a reduction in cholesterol production by the liver, as well as enhancing the uptake of LDL by liver receptors, in turn lowering LDL particle numbers in blood circulation [7]. In a National Health and Nutrition Examination Survey in 2011–2012, 27% of adults aged 40 and overused a cholesterol-lowering medication, with 23% using a statin alone [8].

Fibrates are peroxisome proliferator-activated receptor (PPAR)-alpha ligands. They stimulate the hydrolysis of triglycerides carried by lipoproteins and reduce the circulation time of triglyceride-rich particles, reducing the amount of oxidation and inflammation via PPAR-α-mediated mechanisms that lead to fatty acid streaks and atherosclerosis [9,10]. Fibrates have been historically co-prescribed with statins, as a method of lowering triglycerides, which also contribute to ASCVD risk [11]. Although fibrates are not recommended currently due to side effects and the inhibition of statin metabolism [12], samples collected and archived in the early 2000s may contain fibrates.

The widespread use of lipid-lowering medications creates challenges in designing population studies to assess ASCVD risk due to difficulties in accounting for adherence, which is a concern among health professionals [13]. Broad cross-sectional studies or randomized clinical trials assume patient adherence to medication use and accuracy of medical records. However, medication records are typically collected based on self-reporting through questionnaires or records of prescriptions issued by doctors, which may not accurately reflect the actual use of medication. Due to these challenges, an orthogonal and potentially more accurate way to verify a patient’s medication adherence is by direct analytical measurements of the drugs or their metabolites in plasma. Although methods are available to detect statins and fibrates [14,15,16,17,18,19,20,21,22,23,24,25], many of these methods were developed for the analysis of individual drugs or require volumes of plasma that are often not available from the sample archives. In addition, most analytical methods are not multiplexed enough for effective screening in a large number of samples.

Here, we report the development of a rapid multiplexed method for the simultaneous detection of seven statins and three fibrates (Figure 1) and their metabolites in 10 µL of plasma. The applicability of the method is demonstrated by the analysis of 941 archived plasma samples collected from patients with available records on medication use, allowing comparison with liquid chromatography–mass spectrometry (LC-MS/MS) analysis results.

## 2. Results

### 2.1. Optimization of LC-MS/MS Conditions

All analyte precursors were carboxyl derivatives. For the parent drugs that were esters (simvastatin, lovastatin, fenofibrate, and clofibrate), only the carboxylic acid metabolites gave sufficient S/N. The selected precursor and product ions, declustering potentials (DP), collision energies (CE), and collision cell exit potentials (CXP) were all optimized for each individual compound and are summarized in Table 1.

To maximize the dwell time, the number of points, and the signal-to-noise ratios (S/N) across chromatographic peaks, scheduled multiple reaction monitoring (SMRM) mode was used with ±0.5 min windows around expected retention times. Both negative and positive ion modes were assessed. Negative ionization was selected for a better S/N. The most intense fragments were typically the loss of sidechains and the monitored product ions contained core ring moieties. By adding the same amount of isotopically labeled IS before protein precipitation, the SMRM signal area ratios were sufficiently corrected for variations in the sample preparation recovery and were proportionate to the plasma concentrations.

The reconstitution solvent composition and the eluent gradient program were optimized for minimum injection solvent effects and maximum peak separation in the short 4-min method run time. A two-step gradient that included an initial shallow 0–30% B increase for focusing and an isocratic 60% B period for elution followed by column regeneration provided ample chromatographic resolution to allow SMRM detection while keeping a short 4-min run time. A representative SMRM chromatogram for the IS analogs using optimal chromatographic conditions is shown in Figure 2, and individual analyte chromatograms are shown in Appendix A.

### 2.2. Selection of Protein Precipitation Solvent and Recovery

Percent spike recovery and reproducibility were the main criteria for the selection of the precipitation solvent. The extraction recovery using an 8-fold volume of different solvents (acetone, methanol, and acetonitrile) was evaluated based on absolute LC-MS signal intensities from the spiked plasma relative to the spiked water samples. The highest recoveries with the lowest percent coefficient of variations (%CV) were achieved using acetone (Appendix A). An additional advantage of using acetone was the short and reproducible evaporation in 20 min.

### 2.3. Determination of Detection Criteria

The selectivity was assured by the optimal chromatographic resolution and compound specific precursor/product ion masses. To minimize both false positive and false negative detection of statins and fibrates in plasma, the lower limit of detection (LLOD) was carefully determined. For the initial assessment of the LLOD, 10 µL pooled plasma samples were analyzed at a series of spiked concentrations, 0.01–100 ng/mL, in five replicate extractions. LLOD was the concentration where the native analyte signal intensity was three times above the S/N level. At the selected LLOD level, the CVs of the calculated concentrations were <37% (Table 2).

False positive detection above the method LLOD can occur for a variety of reasons, such as interferences and carryover. We selected six plasma samples that were collected from individuals without statin use and did not show interferences. We assessed the method sensitivity at three spiked concentration levels, at the method LLOD (3× S/N), 2× LLOD (6× S/N), and 5× LLOD (15× S/N), by spiking the six individual plasma samples and their pool (Table 2). All spiked plasma samples were extracted in triplicate. Using the spiked pool extracts as calibrators, the intra-day spiked recovery (accuracy) was 80–120%. The intra-day CV was <30% at the 2× LLOD level and <33% at the 5× LLOD level for all analytes (Table 2).

We compared the measured concentration distributions in the blank and spiked 1× LLOD, 2× LLOD, and 5× LLOD samples. Measured concentrations overlapped only between the 1× LLOD and the 2× LLOD levels (Figure 3), and there was no overlap in concentrations between the 2× LLOD and 5× LLOD levels. Therefore, the 2× LLOD level was chosen as the LOD criterion to evaluate analyte detectability in unknown samples.

As an evaluation of the false positive and false negative detection rates, we performed a differential analysis using the 1× LLOD and 2× LLOD data. The percentage of the number of measured concentrations in 1× LLOD spiked samples that were in the range of the 2× LLOD samples represented false positive detects, and it was 5–20% for all analytes. The percentage of the number of measured concentrations in the 2× LLOD spiked samples that were in the range of the 1× LLOD samples represented false negative detects, and it was 9–20% for all analytes, except for rosuvastatin and clofibrate, for which it was 47%. During the analysis of the unknown samples, we observed >3000 ng/mL levels for fibrates, which, even at a minimal 0.1–0.2% carryover, caused detectable signals in the first consecutive sample. Thus, to avoid false positive results, the detection criterion for fibrates was raised to 30× LLOD for clofibric acid and fenofibric acid, and 20× LLOD for gemfibrozil.

We used deuterium-labeled compounds as internal standards that inherently had some small but reproducible deviation in retention time from the corresponding native analytes. We took advantage of these analyte specific retention time differences (ΔRT) between native and labeled analogs and characterized their mean and standard deviation (Std Dev) using spiked plasma samples (Table 2). We selected ±3× Std Dev around the mean ΔRT as an additional detection criterion for each native analyte at >LOD concentrations.

### 2.4. Application

The self-reported medication use among study participants was 37% (351/941). This group was labeled as Statin-Record [+] (Table 3) and mainly comprised atorvastatin and simvastatin users. When applying our method with the established LOD and ΔRT screening criteria, among participants who reported any statin medication use, we detected a statin compound for 79% (276/351), labeled as Statin-Detect [+]. In the Statin-Record [+] group, the percent match between the detected statin compound and the recorded generic statin medication was 72% (255/341), for atorvastatin 89% (155/176), and for simvastatin 58% (61/105). These percent agreements between medical records and our LC-MS/MS analysis confirmed the sufficient sensitivity and specificity of our method.

The medical records did not indicate any statin use for 62.7% of the participants, labeled Statin-Record [−] (Table 3), and the records did not indicate whether this was due to missing information or mistaken self-reports by patients. In the Statin-Record [−] group, we found detectable statin levels for 42% (250/590) of patients, mainly atorvastatin (176) and simvastatin (46). The mean, median, and range of concentrations for atorvastatin and simvastatin were comparable in the Statin-Record [+] and in the Statin-Record [−] groups (Appendix A). Across atorvastatin-positive samples, the two 2-OH and 4-OH metabolites of atorvastatin showed significant correlations with each other r = 0.629, and with atorvastatin, r = 0.596 and r = 0.506, respectively. The correlations were in the range of r = 0.42–0.60 for Statin-Record [+] samples and r = 0.66–0.73 for Statin-Record [−] samples (Appendix A). Collectively, the similar concentration distributions and the correlation of metabolite concentrations in the Statin-Record [+] and Statin-Record [−] groups provided strong evidence that the LC-MS/MS analysis was likely correct, and the medical records did not reflect statin use accurately.

Fibrate use was not part of the medication recording, and we could not compare with Fibrate-Detect [+]/[−] as for statins. Fibrates were detected in 6.8% and 7.8% of the Statin-Record [+] and [−] groups, respectively. Gemfibrozil and fenofibrate were detected in patient samples, but not clofibrate. Clofibrate was removed from the market in 2002, before the time of the collection of the blood samples. Thus, the non-detection of clofibrate suggests that our LOD and ΔRT criteria for the detection of fibrates were acceptable.

False negative detects in the unknown samples were most likely between the 1× LLOD and 2× LLOD concentration levels. Of all measurable concentrations by each analyte, concentrations between 1× LLOD and 2× LLOD were highest for simvastatin (17%) and rosuvastatin (21%) and lowest for atorvastatin (8%) (Appendix A). However, as shown above, for the spiked samples between 1× LLOD and 2× LLOD levels, both the false positive and negative rates were <20%. The false positive and negative detects in the unknown samples were also proportionally less for simvastatin (~3.4%), rosuvastatin (~4.2%), and atorvastatin (~1.6%). The low chance of false negative detection of atorvastatin and simvastatin was especially low, considering the similarly high median concentrations in Statin-Record [−] and Statin-Record [+] samples, 116 and 86 ng/mL atorvastatin, and 7.2 and 6.2 ng/mL simvastatin, respectively, much higher than the LOD (Appendix A).

## 3. Discussion

### 3.1. Comparing Sensitivity and Accuracy of Our Method with Existing Methods

The LC-MS/MS method developed herein was designed to achieve an optimal balance between a minimal number of false positive and false negative detections while considering speed, multiplexed detection, and limited specimen volume. The total run time was brought down to 4 min by using an ultra-high performance liquid chromatography (UHPLC) platform, which allowed faster analyte elution. Compared to other methods with similar throughput, our workflow detects a greater diversity of statins, fibrates, and their metabolites. The LODs for this method were 0.1–1 ng/mL while using only 10 µL of plasma. In comparison, most literature methods achieve similar LODs while requiring solid phase extraction (SPE), liquid–liquid extraction (LLE), or other sample manipulation techniques, which our described method avoids. Additionally, other methods described in the literature use high volumes of plasma (100–1000 µL) while detecting only 1–7 statins or fibrates [14,15,16,17,18,20,22,23,24,25].

The potential affect of degradation on accuracy and precision has been addressed in previous studies for multiple statins and fibrates in frozen plasma during time frames ranging from 8 days to 2 years at −70 °C and −80 °C. All sources conclude that drugs are stable with detections ranging within +/−15% of the spiked concentrations [17,18,19,21,22,23,25]. Therefore, we assumed stability during long-term storage at −70 to −80°, as our archived samples did not cause significant degradation.

### 3.2. Confidence in the Reported Number of Detects

The analysis of 941 patient samples provides confidence in the applicability of our method for several reasons. Firstly, we confirmed the use of atorvastatin with the highest concordance of 89%, while atorvastatin was the most frequently prescribed generic medication with a long 14-h half-life. Statin use in general was confirmed in 72% of the samples with recorded statin medications. Secondly, we did not detect clofibrate, which has a long 15-h half-life but was not available during the years of the blood collection phase of the study. Thirdly, the amount of atorvastatin and simvastatin detected in patients’ serum was typically at levels far above the LLOD (near the mean or median concentration levels) and rarely near the LLOD level. Fourthly, the number of samples with other detectable statin medications in addition to the most concentrated medication was only 1.2%. Fifth, in the group of samples with and without recorded medications, the mean, median, and standard deviation of the concentrations were similar when considering the two most prescribed generic medications atorvastatin and simvastatin. Sixth, the concentrations of the 2-OH and 4-OH metabolites for atorvastatin strongly correlated with each other and with atorvastatin.

Detectability can be potentially affected by the half-life of the drugs and their metabolites. As an example, simvastatin, which has a relatively short half-life of 2 h, was still detected after around 0.1 ng/mL in 24 h after a 20 mg dose [26]. Our method with a 0.5 ng/mL LLOD did not detect any positive simvastatin sample concentrations below 2 ng/mL (Appendix A).

### 3.3. Accuracy of Statin Use Information Based on Self-Reported Records

Of all individuals, 26.6% (250/941) did not have records of statin use but had detectable statin levels. We found this substantial percentage of unrecorded statin usage in spite of many of the medications having <5 h of half-life in plasma (simvastatin, lovastatin, fluvastatin, pravastatin, and gemfibrozil) [27,28,29,30]. The reason for the unrecorded statin usage may be due to the circumstances of the sample collection. The samples were collected immediately before cardiac catheterization and many individuals who received the referral for this examination were from an elderly age group.

Due to the challenging clinical circumstances of the blood collection, the time when patients took their last medications and their prescribed dosage could not be recorded or controlled. Thus, the measured relative concentrations across individuals could not provide usable information about dosage or adherence to medication use. Therefore, our method was designed with less emphasis on absolute quantification accuracy and precision. Rather, we focused on the ability to maximize the number of detections and matching the medication records by generic statin name, while minimizing the false positive detection of a statin level near the detection limit.

Nonetheless, our results raise concerns about the accuracy of statin use based on self-reports for epidemiologic studies in general, and especially biomarker discovery research. For the discovery of lipid metabolism-related biomarkers, it is a common practice to rank biomarkers based on associations with disease outcomes after correction for statin use with a yes/no binary variable or categorization into user and non-user groups. Regardless, the accuracy of the statin use information may substantially affect both the relative magnitude and the significance of the association of lipid-metabolism-related biomarkers with ASCVD outcomes.

Moving forward, we believe that the lack of concordance between medical records and analysis carries usable information about the study participants’ adherence to medication use. For biomarker discovery work or diet and lifestyle intervention studies, stratification or statistical correction may be useful using a three-level variable, (0): Statin-Record [−]/Statin-Detect [−], (1): Statin-Record [+]/Statin-Detect [−] or Statin-Record [−]/Statin-Detect [+], and (2): Statin-Record [+]/Statin-Detect [+]; or a four-level variable, (0): Statin-Record [−]/Statin-Detect [−], (1): Statin-Record [+]/Statin-Detect [−], (2): Statin-Record [−]/Statin-Detect [+], and (3): Statin-Record [+]/Statin-Detect [+]. Assessing lipid levels by these adherence categories should correspond with decreasing LDL-cholesterol-related lipid profile measures. We plan to present the test of these multi-level correction approaches for the assessment of the association of lipid-metabolism-related biomarkers with ASCVD outcomes in a future publication.

## 4. Materials and Methods

### 4.1. Chemicals and Reagents

Acetone, acetonitrile, propylene glycol, methanol, water, and HPLC-grade acetic acid were purchased from Fisher Scientific (Waltham, MA, USA). Pravastatin-d3, pitavastatin-d5, rosuvastatin-d3, fluvastatin-d6, and 4-OH-atorvastatin-d5 were purchased from BOC Sciences (Shirley, NY, USA). Lovastatin, fenofibrate, fenofibric acid, atorvastatin, simvastatin, simvastatin acid (tenivastatin), rosuvastatin, clofibrate, clofibric acid, and gemfibrozil were purchased from Millipore Sigma (St. Louis, MO, USA). N-desmethyl rosuvastatin was purchased from Santa Cruz Biotechnology (Dallas, TX, USA). Pravastatin, lovastatin acid, 4-OH-atorvastatin, 2-OH-atorvastatin, pitavastatin, fluvastatin, simvastatin-d6, and atorvastatin-d5 were purchased from Cayman Chemicals (Ann Arbor, MI, USA). 2-OH-atorvastatin-d5 and N-desmethyl-rosuvastatin-d3 were purchased from TRC Canada (Toronto, ON, Canada). Clofibrate-d4, clofibric acid-d4, fenofibrate-d6, fenofibric acid-d6, and gemfibrozil-d6 were purchased from CDN Isotopes (Pointe-Claire, Quebec, QC, Canada).

### 4.2. Sample Collection

Deidentified specimens (*N* = 941) from the CATHeterization GENetics (CATHGEN) biorepository, collected by Duke University during 2004–2007, were evaluated in this study (ref PMID: 26271459). Subjects consented to the study in a protocol approved by the Duke University Institutional Review Board. Samples were collected from subjects in a fasting state at the time of coronary artery catheterization. Samples were collected in EDTA-containing sample tubes, spun, and the plasma samples were stored at a cryogenic temperature until 2018. Samples were then shipped on dry ice and stored at −80 °C until use. Prior to statin and fibrate analysis, all samples were thawed on ice less than three times. Along with the samples, deidentified medical records were received containing information about statin use recorded at the time of the blood draw based on self-reports from individual donors. For method development, six blank plasma samples were purchased from BioIVT (Westbury, NY, USA), including three male and three female individuals without any known cholesterol-modifying drug use. These six plasma samples were pooled and used as a matrix for preparing spiked calibrators.

### 4.3. Calibrators, Quality Control Samples, and Internal Standards and Solution

Stock solutions for each native analyte were prepared at 1 mg/mL in methanol, except for rosuvastatin, which required the use of 30:70 MeOH:propylene glycol as a diluent because of the insolubility of rosuvastatin in pure methanol. Deuterium-labeled (IS) stock solutions were also prepared in methanol at 1 mg/mL for each analyte. During method development, different organic solvents containing IS were compared for extraction recovery (Appendix A). For the final optimized method, acetone was used with 10 ng/mL of clofibrate, lovastatin, N-desmethyl rosuvastatin, pravastatin, and simvastatin, 50 ng/mL of fenofibrate, and 2.5 ng/mL of the other internal standards. The structures of the statins and monitored metabolites are shown in Figure 1 and Appendix A [31,32].

### 4.4. Sample Preparation

The method was optimized for the analysis of 10 µL of plasma. Each plasma sample was mixed with 10 µL of water and then 80 µL of acetone spiked with deuterated IS, followed by centrifugation at 14,000 rpm for 10 min. The supernatant was transferred into a semi-skirted conical-bottom 96-well polypropylene microtiter plate and evaporated under a nitrogen flow of 22 L/min at 60 °C for 20 min. Reconstitution of the dry samples was done in the same plate with 25 µL of 30:70 acetonitrile:water.

### 4.5. Liquid Chromatography–Mass Spectrometry Method

Reconstituted samples were analyzed directly via LC-MS/MS in negative ion mode with the SMRM, using ±0.5 min windows around expected retention times. The SMRM table is reported in Table 1. Samples were analyzed by liquid chromatography on a Shimadzu LC-30AD system coupled to a triple quadrupole mass spectrometer (QTRAP5500, AB SCIEX, Framingham, MA, USA). A 15 µL volume of each extract was injected into the LC-MS/MS system. Electrospray ionization was performed with the following optimized source parameters: curtain gas, 20 psi; ionization spray voltage, −4500 V; probe temperature, 450 °C; ion source gas 1, 50 psi; ion source gas 2, 50 psi; entrance potential, −10 V. Chromatographic separation was achieved on a Kinetex C18 (2.1 × 100 mm, 1.7 µm) reverse-phase column at a flow rate of 0.6 mL/min at 40 °C during a 4-min gradient. Mobile phase A contained 0.1% acetic acid in water and mobile phase B contained 0.1% acetic acid in acetonitrile. The solvent gradient at a flow rate of 0.6 mL/min started with 30% B, increased to 60% B in 0.7 min, held constant for 1.4 min, then increased to 95% in 0.2 min, held constant for 0.6 min, reduced to 30% B in 0.1 min, followed by equilibration at 30% B for 1 min.

### 4.6. Data Analysis

Mass spectrometer data were processed and analyzed using the MultiQuant software (AB SCIEX, Framingham, MA, USA). The JMP software (SAS Institute, Cary, NC, USA) was used to determine detectable concentrations.

## 5. Conclusions

In conclusion, we reported the application of a high-throughput multiplex LC-MS/MS method for the simultaneous detection and identification of seven statins and three fibrates in a small volume of human plasma. The method is highly applicable in population studies designed for biomarker discovery or diet and lifestyle intervention studies, where the accuracy of statin or fibrate use may strongly affect the statistical evaluation of biomarker data.

## Figures and Tables

**Figure 1 ijms-24-07931-f001:**
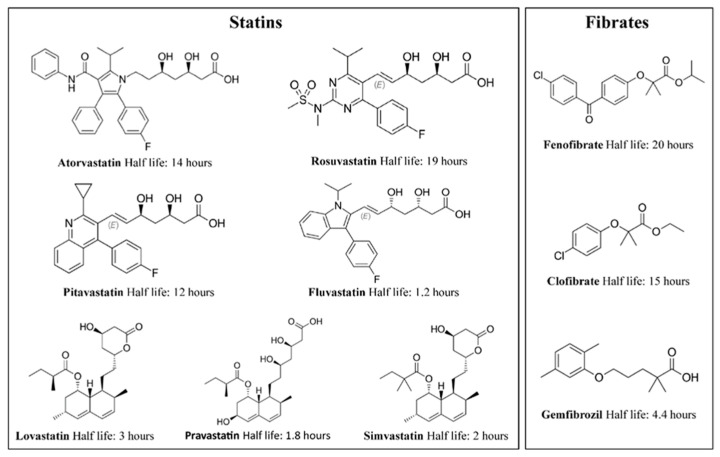
Chemical structures and half-lives of the statins and fibrates. Commercially available statins: lovastatin (Mevacor, Merck Frosst, Toronto, ON, Canada), pravastatin (Pravachol, Bristol-Meyers Squibb, Atlanta, GA, USA), simvastatin (Zocor, Merck Frosst, Toronto, ON, Canada), fluvastatin (Lescol, Novartis, Basel, Switzerland), atorvastatin (Lipitor, Parke-Davis, Detroit, MI, USA), rosuvastatin (Crestor, Astrazeneca, Wilmington, DE, USA), and pitavastatin (LIVALO, Kowa CO, Nagoya, Japan ). Commercially available fibrates: gemfibrozil (Lopid, Pfizer Inc, New York City, NY, USA), clofibrate (Atromid-S, Wyeth, Madison, NJ, USA (discontinued)), and fenofibrate (Tricor, Abbot Laboratories, Abbot Park, IL, USA). Chemical structures for metabolites and fragments can be found in Appendix A.

**Figure 2 ijms-24-07931-f002:**
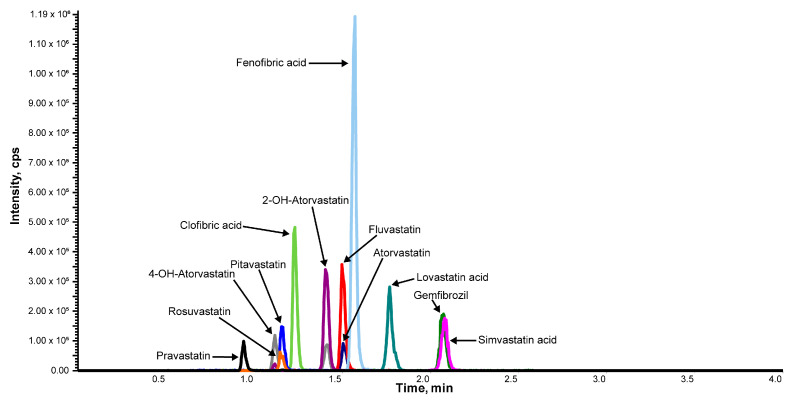
A representative SMRM chromatogram of statins and fibrates. Chromatogram represents IS analogs at optimal chromatographic conditions at 2.5 ng/mL. Each transition was monitored in negative ion mode.

**Figure 3 ijms-24-07931-f003:**
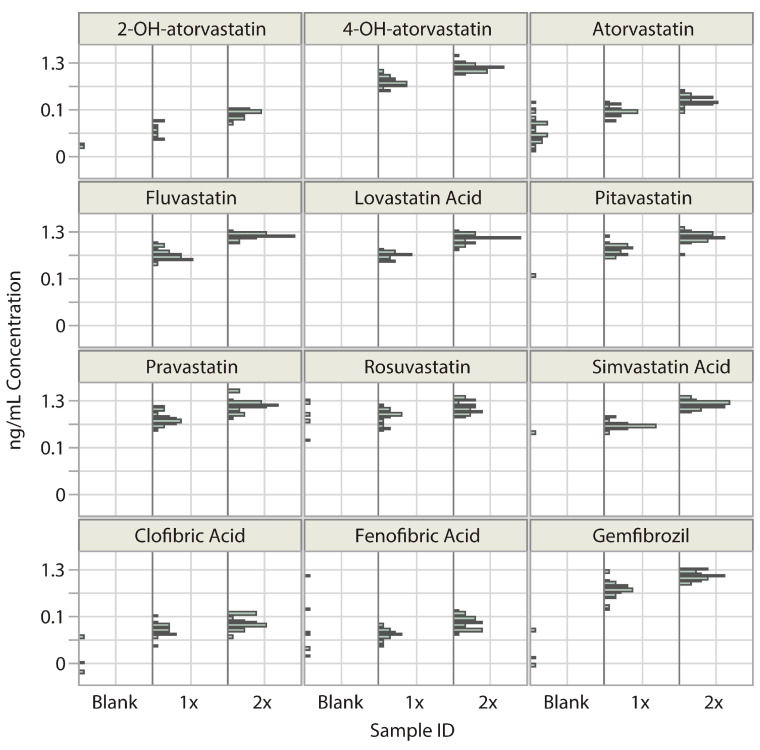
Positive detects in spiked plasma vs. blank of native drugs. Concentration in ng/mL on a log scale in blank, 1× LLOD, and 2× LLOD samples. Detects in 1× LLOD samples in the concentration range of 2× LLOD samples were considered false positive detects. Detects in 2× LLOD in the concentration range of 1× LLOD represented false negative detects.

**Table 1 ijms-24-07931-t001:** Mass spectrometry parameters for targeted analytes. Scheduled multiple reaction monitoring (SMRM) transitions and conditions for deuterated internal standard (IS) and analytes for negative ion mode in ±0.5 min windows around the expected retention time (RT).

Target	Type of the Target	Precursor Mass (Da)	Product Mass (Da)	DP (V)	CE (V)	CXP (V)	RT (min)
2-OH-Atorvastatin	Metabolite of Atorvastatin	573.2	278.0	−73	−39	−33	1.38
2-OH-Atorvastatin-d5	Internal Standard	578.3	283.1	−73	−39	−33	1.38
4-OH-Atorvastatin	Metabolite of Atorvastatin	573.2	413.1	−73	−39	−33	1.09
4-OH-Atorvastatin-d5	Internal Standard	578.2	418.2	−73	−39	−33	1.09
Atorvastatin	Parent Drug	557.2	278.1	−155	−60	−5	1.48
Atorvastatin-d5	Internal Standard	562.1	283.2	−155	−60	−5	1.47
Clofibric Acid	Metabolite of Clofibrate	213.0	127.0	−40	−25	−8.4	1.21
Clofibric Acid-d4	Internal Standard	217.1	131.0	−40	−25	−8.4	1.19
Fenofibric Acid	Metabolite of Fenofibrate	317.1	231.0	−50	−37	−15.6	1.56
Fenofibric Acid-d5	Internal Standard	323.1	231.0	−50	−37	−15.6	1.55
Fluvastatin	Parent Drug	410.3	348.2	−22	−21	−25	1.48
Fluvastatin-d6	Internal Standard	416.2	354.2	−22	−21	−25	1.48
Gemfibrozil	Parent Drug	249.0	121.1	−24	−23	−10	2.09
Gemfibrozil-d6	Internal Standard	255.2	121.1	−24	−23	−10	2.08
Lovastatin Acid	Metabolite of Lovastatin	421.4	101.0	−72	−27	−10	1.75
Lovastatin Acid-d3	Internal Standard	424.3	104.1	−72	−27	−10	1.73
N-Desmethyl-Rosuvastatin *	Metabolite of Rosuvastatin	466.2	404.1	−28	−23	−27	0.85
Pitavastatin	Parent Drug	420.2	358.1	−40	−18	−16	1.13
Pitavastatin-d5	Internal Standard	425.2	363.1	−40	−18	−16	1.12
Pravastatin	Parent Drug	423.4	321.2	−80	−23	−51	0.92
Pravastatin-d3	Internal Standard	426.2	321.2	−80	−23	−51	0.91
Rosuvastatin	Parent Drug	479.8	418.3	−34	−17	−23	1.13
Rosuvastatin-d3	Internal Standard	483.2	421.2	−34	−17	−23	1.12
Simvastatin Acid	Metabolite of Simvastatin	434.9	319.2	−118	−21	−21.4	2.09
Simvastatin Acid-d6	Internal Standard	441.4	319.2	−118	−21	−21.4	2.05

Abbreviations in table: DP: declustering potential; CE: collision energy, and CXP: collision cell exit potential. * Added to method during development, but not in the final method.

**Table 2 ijms-24-07931-t002:** Limit of detection and reproducibility assessment. Relative recovery (%Acc), LLOD, corresponding intra-day and inter-day %CVs, and tolerance range for peak retention time difference (ΔRT) between native and deuterium-labeled analogs, at three concentration levels of spiked blank plasma.

		Spiking Levels (ng/mL)											
		1× LLOD				2× LLOD				5× LLOD			
Analytes	LLOD (ng/mL)	Intra-Day (*n* = 18)		Inter-Day (*n* = 24)		Intra-Day (*n* = 18)		Inter-Day (*n* = 24)		Intra-Day (*n* = 18)		Inter-Day (*n* = 24)	
		%Acc	%CV	%CV	ΔRT	%Acc	%CV	%CV	ΔRT	%Acc	%CV	%CV	ΔRT
2-OH-Atorvastatin	0.05	106	26	36	0.026	106	13	36	0.025	96	8	33	0.040
4-OH-Atorvastatin	0.5	84	13	20	0.019	101	11	20	0.018	107	13	29	0.017
Atorvastatin	0.1	97	25	37	0.058	94	26	37	0.037	88	18	30	0.043
Clofibric Acid	0.05	140	37	38	0.064	102	22	38	0.069	100	19	23	0.073
Fenofibric Acid	0.05	127	31	25	0.046	120	20	25	0.044	113	10	21	0.140
Fluvastatin	0.5	117	13	21	0.022	110	12	21	0.016	97	9	15	0.018
Gemfibrozil	0.5	119	5	34	0.019	113	4	34	0.019	105	7	9	0.017
Lovastatin Acid	0.5	99	13	33	0.024	98	15	33	0.024	97	7	22	0.023
Pitavastatin	0.5	117	30	29	0.013	108	27	29	0.016	104	7	17	0.018
Pravastatin	0.5	84	13	44	0.015	95	12	44	0.013	101	8	9	0.010
Rosuvastatin	0.5	140	29	47	0.027	105	28	47	0.027	113	14	24	0.024
Simvastatin Acid	0.5	100	11	37	0.030	106	5	37	0.019	104	6	14	0.026

**Table 3 ijms-24-07931-t003:** Comparison of statin use based on medical records and LC-MS/MS analysis.

Recorded Medication	Ator	Sim	Pra	Lov	Rosu	Flu	Sim/Eze	Statin-Record [+]	Statin-Record [−]
Medication Statin Record	174	105	33	12	11	3	5	Σ351	590
(% Statin-Record [+]/Σ941	(19.0)	(11.4)	(3.6)	(1.3)	(1.2)	(0.3)	(0.5)	(37.3)	(62.7)
% of Statin-Record [+]/Σ351	49.6	29.9	9.4	3.4	3.1	0.9	1.4		
**Analyzed Statins**	Number of samples with recorded statin medication							Σ Statin-Detect [+] in Statin-Record [+]	Σ Statin-Detect [+] in Statin-Record [−]
Atorvastatin [+]	155	9	3	2	0	0	1	170	176
Simvastatin [+]	1	61	1	0	1	0	2	66	46
Pravastatin [+]	0	0	20	0	0	0	0	20	5
Lovastatin [+]	0	0	0	8	0	0	0	8	6
Rosuvastatin [+]	0	2	0	0	6	0	0	8	15
Fluvastatin [+]	1	1	0	0	0	2	0	4	2
Pitavastatin [+]	0	0	0	0	0	0	0	0	0
Σ Statin Detect [+]	Σ157/174	Σ73/105	Σ24/33	Σ10/12	Σ7/11	Σ2/3	Σ3/5	Σ276/Σ351	Σ250/590
% Statin Detect [+] confirmed	90.2	69.5	72.7	83.3	63.6	66.7	60.0	78.6	42.4
Medication confirmed	155/174	61/105	20/33	8/12	6/11	2/3	2/5	Σ255/351	
% Medication confirmed	89.1	58.1	60.6	66.7	54.5	66.7	40.0	72.4	
**Analyzed Fibrates**	Number of samples with recorded statin medication							Σ Fibrate-Detect [+] in Statin-Record [+]	Σ Fibrate-Detect [+] in Statin-Record [−]
Clofibrate [+]	0	0	0	0	0	0	0	0	0
Fenofibrate [+]	9	4	1	0	0	0	0	14	23
Gemfibrozil [+]	6	2	2	0	0	0	0	10	23
Σ Fibrate Detect [+]	Σ15/174	Σ6/105	Σ3/33	0	0	0	0	Σ24/Σ351	Σ46/590
% Fibrate Detect [+]	8.6	5.7	9.1	0	0	0	0	6.8	7.8

Abbreviations: Ator: Atorvastatin, Sim: Simvastatin, Pra: Pravastatin, Lov: Lovastatin, Rosu: Rosuvastatin, Flu: Fluvastatin, Sim/Eze: Simvastatin/Ezetimibe.

## Data Availability

The datasets used and/or analyzed during the current study are contained within the manuscript and available from the corresponding author on reasonable request.

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
