# Peer review of "Confirmation of Statin and Fibrate Use from Small-Volume Archived Plasma Samples by High-Throughput LC-MS/MS Method"

_ijms, 2023, doi:10.3390/ijms24097931_

Round 1
Reviewer 1 Report
Dear Authors,
The manuscript submitted to me for review meets all the criteria for a good, intelligent and highly scientific analytical work in the field of drug monitoring.
This MS report the development of a rapid multiplexed method for the simultaneous detection of 7 statins and 3 fibrates and their metabolites in a minimal volume (10 µL) of plasma. The applicability of the method is demonstrated by analysis of 941 archived plasma samples collected from patients with available medication use records, allowing comparison with the results of liquid chromatography-mass spectrometry (LC-MS/MS) analysis.
The work is within the scope of the journal. The MS is original, well-defined and reports results that have not been submitted or published before. The title is concise, synthetic and informative.
The results are significant, appropriately interpreted and provide advances in the knowledge of drug analysis in biological fluids such as plasma. The figures and tables are comprehensive, correctly present the data, and are easy to understand. The study is very well designed and technically sound. The analyses were performed with the highest technical standards in the field of analytical chemistry and biochemistry. Highly informative methods of analysis, statistical processing and presentation of results were also used. The data are sufficiently robust to draw conclusions. The methods, instruments, software and reagents are described in sufficient detail to allow other researchers to reproduce the results. Additional data and results from the analyses and measurements are also available. Conclusions are justified, supported by the results, and of interest to readers of the journal.
The paper is written in an appropriate manner. The English language is good and understandable.
Author Response
We thank you for the insightful and positive comments on our manuscript. We appreciate that you found this study “very well designed and technically sound” and “intelligent and highly scientific analytical work in the field of drug monitoring”.

Reviewer 2 Report
This manuscript describes a methodology to evaluate the use of statins and fibrates in samples which are meant for lipidomics research. The method offers high throughput and is able to evaluate the presence and estimate the concentration of several molecules of the mentioned classes of molecules.
Introduction is well constructed; maybe some more accent could be put on the limitations of current standard in lipid profiling. In general the manuscript is well-written, results presented clearly and the discussion adequate. From my perspective this study should be accepted for publication after including in the discussion the following aspects:
1. Is there evidence in literature regarding the stability of statins and fibrates in plasma samples over long periods of time? It would be interesting to evaluate whether the absence of statins and fibrates in some samples could be due to stability issues or patient not taking their treatment.
2. How can these results be translated into better lipidomics studies from a practical point of view? Exclude the patients which received this treatment from the lipidomics study? Correct the data in some way?
Other observations:
line 311-312: "Samples were collected by subjects..." can be misinterpreted. Please reformulate.
line 29: "statin" instead of "stain"
line 338: I presume LPM represents liters per minute. Please replace LPM with L/min or equivalent.
Author Response
- Is there evidence in literature regarding the stability of statins and fibrates in plasma samples over long periods of time? It would be interesting to evaluate whether the absence of statins and fibrates in some samples could be due to stability issues or patient not taking their treatment.
Authors’ response: In the manuscript we already included 6 references and we added an additional reference on statin stability, including those that assessed from 8 days to 2 years for various statins and fibrates. The reference publications indicated no significant degradation of parent drugs and metabolites within 85-115% of initial concentration recovery. This supports drug stability in archived samples stored at cryogenic conditions. The manuscript has been edited to contain this information. See Lines 244-249 and 266-271.
- How can these results be translated into better lipidomics studies from a practical point of view? Exclude the patients which received this treatment from the lipidomics study? Correct the data in some way?
Authors response: We agree with the reviewer that having accurate statin use data is important because of the powerful lipid lowering effects of statins. We are currently looking into statistical methods to best deal with this issue for assessment of the association of lipid metabolism-related biomarkers with ASCVD outcomes in our future publication. This method will be a powerful tool to help resolve these issues. Please see line 296-309 in the manuscript.
- Other observations and corrections:
line 311-312: "Samples were collected by subjects..." This statement was modified for clarity.
line 29: Corrected spelling of "statin”
line 338: Replaced LPM with L/min.
